extreme events; wave transformation; nonlinear waves; environmental fluid mechanics; ocean engineering

**Corresponding authors:**
Amin Chabchoub and Yan Li;
Email: chabchoub.amin.8w@kyoto-u.ac.jp;
Yan.Li@uib.no

# On the formation of coastal rogue waves in water of variable depth

Yan Li[1] [ID] and Amin Chabchoub[2,3,4] [ID]

[1]Department of Mathematics, University of Bergen, Bergen 5020, Norway; [2]Disaster Prevention Research Institute, Uji, Kyoto University, Kyoto 611-0011, Japan; [3]Hakubi Center for Advanced Research, Kyoto University, Yoshida-Honmachi, Kyoto 606-8501, Japan and [4]School of Civil Engineering, The University of Sydney, Sydney, NSW 2006, Australia

## Abstract

Wave transformation is an intrinsic dynamic process in coastal areas. An essential part of this process is the variation of water depth, which plays a dominant role in the propagation features of water waves, including a change in wave amplitude during shoaling and de-shoaling, breaking, celerity variation, refraction and diffraction processes. Fundamental theoretical studies have revolved around the development of analytical frameworks to accurately describe such shoaling processes and wave group hydrodynamics in the transition between deep- and shallow-water conditions since the 1970s. Very recent pioneering experimental studies in state-of-the-art water wave facilities provided proof of concept validations and improved understanding of the formed extreme waves' physical characteristics and statistics in variable water depth. This review recaps the related most significant theoretical developments and groundbreaking experimental advances, which have particularly thrived over the last decade.

## Impact statement

The fundamental understanding of wave–seabed interactions is crucial for the establishment of accurate extreme wave statistics and deterministic wave prediction in such water-depth-varying zones. With the increase in wind intensities resulting from global warming and respective change in climate dynamics, it is anticipated that the frequency of rogue wave events, occurring in particular offshore and coastal areas, will increase in the future. It is therefore essential to fully understand the formation and complex dynamics of large-amplitude waves in varying water depth conditions, for instance, when deep-water wave groups are transitioning to shallow-water areas. Moreover, quantifying the role of nonlinearity in such wave shoaling or focusing processes is, among other things, decisive for the estimation of associated wave loads on coastal structures and their impact on the shoreline components.

## Background

Ocean water depth is a key parameter in the modeling of waves. In fact, it affects the dispersion relation and characteristic shape features. The change of water depth can be either localized in the form of seamounts and submerged volcanic islands or continuously varying such as continental shelves, encasing nearshore sandbars (Dingemans, 1997; Svendsen, 2005), as exemplified in Figure 1.

Extreme wave formation, being the specific subject of interest of this review article, is not restricted to a particular water depth. In fact, it is known that such large-amplitude waves have been widely reported and recorded not only offshore, but also in coastal zones (Kharif et al., 2008; Dudley et al., 2019; Didenkulova, 2020; Gemmrich and Cicon, 2022). Recent studies suggest that extreme wave conditions are likely to increase as a consequence of climate change (Meucci et al., 2020), even though there are some uncertainties in the modeling and hindcast projections which must be considered (Morim et al., 2023). Regardless of either the superposition principle or wave instability as the underlying focusing mechanism at play, the role of nonlinearity in the wave shoaling transformations is indisputable. The impact of such nonlinear effects is enhanced with the decrease of water depth as a result of Stokes bound harmonics accentuation and contribution to the change of wave shape profile and celerity, see Mei et al. (1989), Osborne (2010), Babanin (2011).

While the role of modulation instability (MI), which is triggered as a result of four-wave quasi-resonant interaction and third-order nonlinear effects, has been intensively studied in deep water for decades (Waseda, 2020), it is only recently that systematic experimental progress has been accomplished in characterizing key statistical features of waves propagation and the role of high-order nonlinear effects when either isolated wave groups or irregular waves propagate atop depth transitions. The presence of MI (Zakharov, 1968) in an irregular wave field can be quantified by

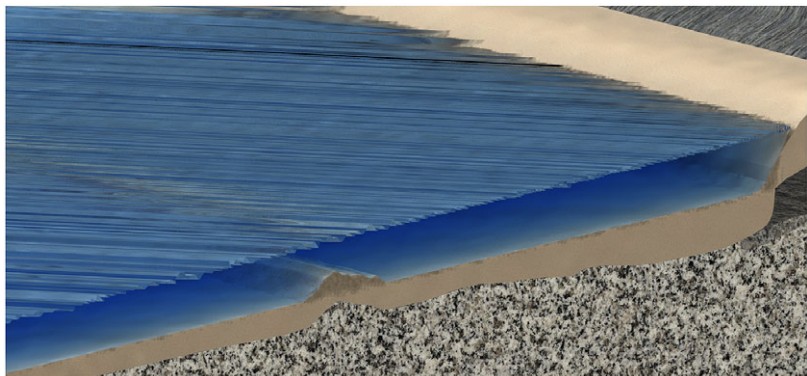

**Figure 1.** A schematic exemplifying typical depth variations from a ridge and continental shelf.

computing the deviation of the surface elevation probability distribution's fourth spectral cumulant, i.e. the kurtosis, from the value of three, which is typical for a Gaussian process (P. A. E. M. Janssen, 2003; Mori et al., 2011).

Such fundamental understanding is not only crucial to improve wave modeling and prediction, but also to better assess wave loads on coastal installations (Li et al., (2023) and provide an accurate nonlinear depth-inversion framework (Martins et al., 2013).

This review paper puts an emphasis on essential recent progress in nonlinear wave modeling and the occurrence of extreme conditions when waves propagate over different types of variable bottom topographies. These advances can be categorized either by the dominance of second-order effects of wave approximation or by the inclusion of third-order contributions to accurately describe the extreme wave dynamics. We will also discuss the essential and complementing laboratory experiments, comprising simplified and complex bathymetries, which have been conducted either for model validation purposes or to drive respective theoretical and numerical progress.

## Physical mechanisms and modeling

Extremely large wave events, also known as rogue, freak or monster waves, are characterized by two main features: their sudden appearance out of nowhere and strikingly amplified steepness compared to their surroundings, thus posing a great risk to the safety and reliability of offshore structures as well as coastal management and protection in nearshore waters (Bitner-Gregersen and Gramstad, 2015). Indeed, there have been documented accidents caused by extreme waves in both intermediate and shallow water (Chien et al., 2002; Didenkulova and Anderson, 2010; Gramstad et al., 2013). In order to characterize rogue wave events, a few useful proxies are commonly used, including skewness and kurtosis which correspond to the third and fourth moment of surface elevation, respectively (Janssen, 2003; Dysthe et al., 2008; Mori et al., 2011). These proxies are used to measure the degree of deviation from the Gaussian random process, thus indicating the occurrence probability of rogue waves.

The properties of surface gravity waves are affected by a seabed in intermediate or shallow water, leading to a complex interplay with wave nonlinearity, compared with deep-water waves. Noticeably, the MI appearing at the third-order in wave steepness approximation has been recognized as a possible mechanism for

the formation of rogue waves in deep water (Benjamin and Feir, 1967; Zakharov, 1968). It can be stabilized for small-amplitude and long-crested waves in water regions for $kh \lesssim 1.36$, where $k$ and $h$ denote the characteristic wavenumber and local water depth, respectively (Johnson, 1977). The threshold value of $kh \approx 1.36$ is essential to the understanding of rogue waves and the evolution of nonlinear energy transfers in finite and shallow water (Janssen and Onorato, 2007). That said, for finite-amplitude waves, Benney and Roskes (1969), McLean (1982), Toffoli et al. (2013) showed that the MI, as a combination of quartet resonant wave interaction and wave nonlinearity, can occur even in water regions where $kh \lesssim 1.36$, when the waves are subject to oblique perturbations. The directional spreading and wave dissipation in random sea states in uniform and finite water depth have been found to lead to considerable deviations from normal statistics (see, e.g., Fernandez et al., 2015; Karmpadakis et al., 2019, among others), while numerical simulations based on the high-order spectral method (HOSM) have indicated the emergence of significant deviations from normal statistics in random directional sea states in the absence of breaking dissipation and independently of the significance of directional spreading of wave spectra in the same uniform depth conditions (Toffoli et al., 2009).

Compared with an intermediate uniform depth, the underlying fundamental physics of surface waves experiencing an additional depth decrease becomes more complex, attributing to the linear refraction and diffraction and the interaction between a varying seabed and wave nonlinearity (Kirby and Dalrymple, 1983; Tsay, 1984; Dingemans, 1997). The coupled effects of wave nonlinearity and a varying bathymetry are the focus of this section.

As noted, wave transformation in variable water regions has been intensively studied in the last decade; these studies are mainly arising from new findings associated with the increased likelihood of extremely large wave events in such alternating depth regions. In their experimental observations, Trulsen et al. (2012) report both a non-homogeneous distribution of skewness and kurtosis of surface displacement and their anomalous behavior in the neighborhood of the water region atop a depth decrease. This suggests an enhanced occurrence probability of rogue waves in this particular region. Similar findings have been reported in other theoretical studies (Li et al., 2021a,b,c), numerical simulations (Sergeeva et al., 2011; Zeng and Trulsen, 2012; Gramstad et al., 2013; Viotti and Dias, 2014; Ducrozet and Gouin, 2017; Lawrence et al., 2021; Lyu et al., 2021) and later-on experimental observations (Ma et al., 2014; Bolles et al., 2019; Kashima and Mori, 2019; Zhang et al., 2019;

Trulsen et al., 2020; Li et al., 2021a,b,c) in a large range of water depth, as will be discussed in the next section. The numerical simulations based on the standard one-dimensional Boussinesq equations, carried out by Gramstad et al. (2013), suggest that the non-homogeneous wave statistics can only be observed for waves propagating from a deeper to shallower water region, but not vice versa. This finding is similar to Armaroli et al. (2020), which concludes that the MI for long-crested waves propagating atop a slowly increased water depth can be stabilized subject to nonlinear evolution, suggesting a possible increase in the lifetime of unstable wave groups, when the water level experiences a depth increase. The features of extreme waves in a varying water region are in principle complex as they are altered by a number of physical parameters such as the non-dimensional wave depth $kh$; the "mildness" of the depth variation relative to the change of wavelength; Ursell number, which measures the degree of the wave nonlinearity relative to a local water depth; directional spreading; the profile shape of a varying bathymetry; and the difference and ratio of water depths (Sergeeva et al., 2011; Zeng and Trulsen, 2012; Viotti and Dias, 2014; Ducrozet and Gouin, 2017; Kashima and Mori, 2019; Zheng et al., 2020; Kimmoun et al., 2021; Li et al., 2021a,b,c; Lawrence et al., 2022). The location where the largest probability of extreme waves atop a varying bathymetry may occur has also been found to coincide with the one where the monochromatic surface waves start to break as the waves steepen (Draycott et al., 2022). Different from the aforementioned findings, the local peak of kurtosis and skewness near the top of a mildly shoaling slope was not reported in Zeng and Trulsen (2012) using numerical simulations, confirmed by Lawrence et al. (2021). This suggests an enhanced number of extreme waves in a varying water region requires the bathymetry to not vary in an extremely mild manner.

A few fundamental physical mechanisms for the formation of extremely large wave events over depth transitions have been proposed in the last decade (Li et al., 2021a,b,c). The second-order nonlinearity dominant mechanisms are first highlighted. In agreement with the second-order dominant physics, as has been pointed out by Gramstad et al. (2013), these are referred to as the processes in which the underlying physics is considered and approximated up to the second-order in wave steepness. A physics-based statistical model is derived by Li et al. (2021a,b,c) based on a deterministic wavepacket model (Foda and Mei, 1981; Massel, 1983; Li et al., 2021a,b,c). As weakly nonlinear waves propagate over an intermediate uniform depth, it has been well known that the waves forced by the second-order nonlinearity are bound (or locked) as they do not obey the linear dispersion relation, see for instance Phillips (1960), Dalzell (1999) and Li and Li (2021) among others. Indeed, a second-order, three-dimensional, finite-depth wave theory can well interpret in-situ measurements of short-crested wind waves, which are observed to cause a setup instead of setdown below large wave groups (Toffoli et al., 2007). In contrast, the statistical model proposed by Li et al. (2021a,b,c) accounts for the complementary physics of the nonlinear forcing of free waves, attributing to the complex interaction between the second-order bound waves and a varying seabed. Both the additional physics and the statistical model by Li et al. (2021a,b,c) have been validated by rigorous theoretical derivations as well as numerical and experimental observations. We refer to Foda and Mei (1981), Massel (1983), Ohyama and Nadaoka (1994), Monsalve Gutiérrez (2017), and Li et al. (2021a,b,c) for more details. The additionally released free waves carry energy and propagate at a different speed from the bound waves responsible for their generation. The differences in the propagation speed of waves lead to their separation at a distance sufficiently far from the top of depth transitions, thus leading to non-homogeneous wave features (Massel, 1983; Li et al., 2021a,b,c; Draycott et al., 2022). We would also like to stress that another physics-based model has been derived by Majda et al. (2019) for shallow-water extreme waves experiencing depth transitions. It is based on truncated Korteweg–de Vries equations and statistical matching conditions of wave fields before and after the depth transition. Both Majda et al. (2019) and Li et al. (2021a,b,c) assume quasi-Gaussian statistics processes for waves on the deeper (constant-water) side of the depth transition.

A second-order statistical non-Gaussian model has been recently derived by Mendes et al. (2022) and Tayfun and Alkhalidi (2020) with respect to the wave heights and free surface elevation, respectively, for waves atop a local intermediate depth transition. Mendes et al. (2022) neglect the second-order subharmonic bound waves, and this finding has been extended in the following work by Mendes and Kasparian (2022) to allow for the effects of a varying seabed slope. In contrast to Li et al. (2021a,b,c), the statistical models of non-Gaussianity neglect the second-order subharmonic (bound and free) waves, the complex interaction between second-order superharmonic bound waves, a varying seabed and the effect of wave reflection. The non-homogeneity of the wave statistical features predicted by the non-Gaussian models originates from a non-constant depth, meaning that the predicted statistical wave features remain invariant with the space if the water is uniform in a local region. This suggests that the model is expected to fail when accounting for the local peaks of skewness and kurtosis near the flat top region of depth transitions. The non-homogeneity of skewness and kurtosis of the surface elevation has particularly been investigated in a number of papers, for example, Trulsen et al. (2012), Zeng and Trulsen (2012), Ducrozet and Gouin (2017) and Zheng et al. (2020).

It shall be noted that the second-order-based mechanisms are in general insufficient for the predictions of kurtosis evolution as the combined effect of the linear waves and third-order nonlinearity cannot be considered. These higher-order effects play a considerable role in the deviation from Gaussian statistics (Janssen, 2014) and are discussed next.

The mechanism of *out-of-equilibrium dynamics* of wave fields has been initially discussed by Viotti and Dias (2014), and thereafter by a number of works, for example, the review by Onorato and Suret (2016) and Trulsen (2018). It is referred to as the process of re-adjusting wave fields from one equilibrium to a new one due to local changes in the environmental conditions, for example, varying bathymetries (Viotti and Dias, 2014; Zhang et al., 2019; Lawrence et al. 2021, 2022; Zhang and Benoit, 2021; Zhang et al., 2023), non-uniform currents (Hjelmervik and Trulsen, 2009; Onorato and Suret, 2016; Zheng et al., 2023) or the sudden appearance of a ship (Molin et al., 2014). The out-of-equilibrium dynamics of wave fields mainly arise from the quasi-resonant wave interaction at third-order in nonlinearity, leading to the deviation of statistical properties of surface elevation from Gaussian statistics (Janssen, 2003; Onorato and Suret, 2016; Tang et al., 2021). This can lead to a change in the spectral bandwidth (Beji and Battjes, 1993), accompanied by a variation of the skewness (Onorato and Suret, 2016), and consequently the occurrence of extreme waves (Viotti and Dias, 2014).

Here, we find the experimental observations by Trulsen et al. (2020), which report different non-homogeneous features of the kurtosis of surface elevation and wave kinematics due to long-crested waves atop a submerged bar allowing for transitions between deep-water and intermediate depths, very instructive.

These findings have a significant impact on the hydrodynamic loads, and therefore also the design of structures in coastal waters (Bitner-Gregersen and Gramstad, 2015; Trulsen et al., 2020; Ghadirian et al., 2023). The experimental results have been confirmed by means of the HOSM (Lawrence et al., 2021). Later, these findings have been extended to account for two-dimensional bathymetry using HOSM-based numerical simulations by Lawrence et al. (2022). So far, no satisfying theoretical explanations have been proposed for the phenomenon reported by Trulsen et al. (2020), although the statistical model of surface elevation by Mendes et al. (2022) and Mendes and Kasparian (2022) can predict non-homogenous statistical features of surface elevation, but are limited to a local water region in which the depth is assumed to vary in space. Especially, whether or not differences between the statistical features of wave kinematics and surface elevation appear in other general contexts is an open question and subject to future studies.

## Experimental investigation

Isolated extreme wave creation in a group atop a changing bathymetry in water wave facilities has attracted the attention of experimentalists in wave hydrodynamics since the 1990s (Baldock and Swan, 1996; Whittaker et al., 2017). The wave group focus has been modeled based on the wave superposition principle while accounting for higher harmonics corrections in the boundary conditions adopted to initiate the experiments (Ma et al., 2022). Such considerations are crucial for the precise wave generation as well as accurate assessments of flow kinematics (Faltinsen et al., 1995; Borthwick et al., 2006), swash oscillations on the beach (Baldock and Holmes, 1999), sediment transport estimates and scour around a pile (Sumer and Fredsøe, 2002; Aagaard et al., 2012), and wave loads on structures (Zang et al., 2010; Ghadirian and Bredmose, 2019; Li et al., 2023).

More recently, experimental studies investigating the effect of bathymetry slope change on either quasi-steady (Li et al., 2021a,b,c) or modulationally unstable wave groups (Kimmoun et al., 2021) confirmed that the role of second-order effects is dominant during the extreme wave group transformation on a slope bathymetry. Having said that, the unstable wave groups did not swiftly demodulate over steep slopes when reaching depth regions $kh < 1.363$, known to be the water regime for the MI to be inactive for unidirectional wave propagation (Johnson, 1977; Mei et al., 1989).

When analyzing more realistic conditions, that is, broadband wave signal of JONSWAP-type representative sea state initialization, as parametrized in Hasselmann et al. (1973), as well as considering the propagating of the respective irregular waves in variable depth

conditions, groundbreaking key findings from laboratory wave data have been reported since the first pioneering study of its kind by Trulsen et al. (2012). In the latter and as mentioned earlier, it has been shown that a local maximum in skewness and kurtosis occurs on the shallower side of a linear slope, suggesting the increase of extreme wave probability in the neighborhood of the top of the depth change. Follow-up studies continued the investigation of the role of nonlinearity in the extreme wave emergence over a variable floor depth while considering a similar unidirectional experimental setup, that is, as utilized and described by Trulsen et al. (2012) atop either a submerged bar or different linear slope inclinations (Kashima et al., 2014; Ma et al., 2014; Kashima and Mori, 2019; Zhang et al., 2019). Schematics of a state-of-the-art apparatus are shown in Figure 2 (a) while (b) shows the corresponding evolution of surface elevation kurtosis as measured from the wave gauges. An excellent progress timeline has been provided by Trulsen et al. (2020) in their Figure 1. It is worth highlighting the work of Kashima and Mori (2019), which suggests that for steep bathymetry slopes, third-order nonlinear effects are still active, even though the dimensionless depth regime $kh < 1.363$ is not supposed to allow the quasi-four waves resonant interactions to unfold – a fact, also confirmed in experiments and numerical simulations by Kimmoun et al. (2021). Moreover, the study by Zhang et al. (2019) emphasized that advanced numerical simulations, such as the Boussinesq-type model, can excellently reproduce the key statistical features of the experiments and confirmed the simulation results of Gramstad et al. (2013).

Follow-up breakthrough contributions discussing the role of the shoal depth and the mismatch of the location of kurtosis peak of the surface elevation and horizontal fluid velocity on the lee side of the shoal have been reported by Trulsen et al. (2020), as already elaborated upon in the previous section, while the results by Li et al. (2021a,b,c) underpinned the generation of new second-order free waves responsible for the wave focusing.

It is also pertinent to note that an abrupt bathymetry change from finite to deep-water conditions can freeze modulationally unstable wave groups to steady packets (Gomel et al., 2021).

An experimental campaign comprising a more sophisticated experimental setup consisting of a submerged bar and an accelerating uniform current revealed that up to a certain shoal depth threshold, the presence of such a flow forcing can enhance the non-Gaussianity of a sea state, thus increasing the frequency of extreme event formation (Zhang et al., 2023).

There are also excellent experimental contributions discussing long-wave focusing and tsunami-type wave shoaling behavior (Goseberg et al., 2013; Pujara et al., 2015), and the occurrence of

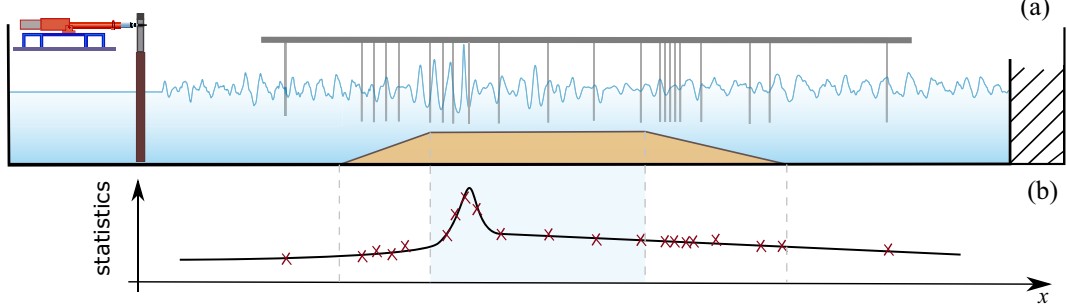

**Figure 2.** (a) Schematic representation of a state-of-the-art experimental setup to study nonlinear wave shoaling and de-shoaling dynamics. The arrangement includes a computer-controlled wave generator, wave gauges, a wave absorber and a submerged bar. (b) Example of the non-homogeneous distribution of statistics (e.g., kurtosis) of surface elevation for random waves atop a submerged bar, as determined from the measurements.

rogue waves in opposing currents (Toffoli et al., 2015). However, these will not be discussed as being beyond the scope of this review.

## Summary and outlook

Our brief review article comprises an overview of the latest physical modeling and experimental validation studies addressing the formation of isolated extreme wave events when transitioning from a deep to a shallow environment, and in some cases the other way around through a specific change in the bathymetry. The progress has been particularly significant and impactful over the last decade, underlining the need of studying such flow dynamics and statistics to confront the global warming–related increase of wind speeds and associated wave heights in the future.

Even though the theoretical, numerical and experimental advances have been "overwhelming", as reported, there are still crucial improvements that have to be made in the modeling. This is to address realistic conditions for different and varying coastal morphologies as well as converging towards common outcomes and conclusions when including directional sea states propagating over a shoal (Bitner, 1980; Cherneva et al., 2005; Ducrozet and Gouin, 2017; Lawrence et al., 2022; Lyu et al., 2023). Whether or not differences between the statistical features of wave kinematics and surface elevation appear in other general contexts and how the differences affect the design standards of coastal structures are open questions for future studies. Moreover, the fast developments of computational capacities will allow the study of this physical problem within the framework of a more advanced numerical framework, such as two-phase flows solving the Navier–Stokes equations or smoothed particle hydrodynamics, applied to realistic domain configurations. Last, but certainly not least, we anticipate that newly developed machine learning algorithms, if fed with high-fidelity data, will play a major role in the operational detection of nearshore extreme waves in the near future.

**Open peer review.** To view the open peer review materials for this article, please visit http://doi.org/10.1017/cft.2023.21.

**Acknowledgments.** The authors thank Yuchen He for assistance in the figure preparation.

**Financial support.** Y.L. acknowledges support from the Research Council of Norway through the POS-ERC project 342480. A.C. is supported by the Hakubi Center for Advanced Research at Kyoto University.

**Competing interest.** The authors declare no competing interests exist.

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
