## [Reviewer Report]

On the formation of coastal extreme waves in water of variable depth

Li and Chabchoub

This is an interesting review manuscript that addresses formation of extreme waves in water of finite uniform depth and, more specifically, over a variable bathymetry. The latter is the primary focus of this work as this bathymetric condition is far more realistic than uniform depth. Overall, the manuscript is well written, comprehensive and suitable for publication. As this is a review, I only have a few minor comments related to some missing literature, which the authors may consider when preparing revised version of the manuscript:

#1: in the last paragraph of page 2, the authors discuss conditions of finite uniform depth. For completeness, it should be mentioned that numerical simulations with the HOS method has indicated emergence of significant deviations from normal statistics in random directional sea states in the absence of breaking dissipation

Toffoli, A., Benoit, M., Onorato, M. and Bitner-Gregersen, E.M., 2009. The effect of third-order nonlinearity on statistical properties of random directional waves in finite depth. Nonlinear Processes in Geophysics, 16(1), pp.131-139.

Conversely, laboratory experiments in similar finite uniform depth have shown that extreme waves do occur more often that in normal statistics primarily due to wave breaking dissipation: see

Fernandez, L., Onorato, M., Monbaliu, J. and Toffoli, A., 2016. Occurrence of extreme waves in finite water depth. Extreme ocean waves, pp.45-62;

Karmpadakis, I., Swan, C. and Christou, M., 2019. Laboratory investigation of crest height statistics in intermediate water depths. Proceedings of the Royal Society A, 475(2229), p.20190183

#2 There is an interesting discussion on second-order theory in the manuscript. I would advise to briefly explain how second order theory compare against field observations in finite depth. Like for modulation instability, second order theory is affected by directional distributions, which introduces a set up under the most energetic groups contributing to amplifying amplitude of the largest waves. Accounting for broad directional distribution, a second order model replicate field data of wind generated waves correctly: see

Toffoli, A., Monbaliu, J., Onorato, M., Osborne, A.R., Babanin, A.V. and Bitner-Gregersen, E., 2007. Second-order theory and setup in surface gravity waves: a comparison with experimental data. Journal of physical oceanography, 37(11), pp.2726-2739.

#3 At page 4, out of equilibrium in non uniform currents is mentioned. I think a relevant reference that should be added is 

Toffoli, A., Waseda, T., Houtani, H., Cavaleri, L., Greaves, D. and Onorato, M., 2015. Rogue waves in opposing currents: an experimental study on deterministic and stochastic wave trains. Journal of Fluid Mechanics, 769, pp.277-297

which show comprehensive experimental evident of the phenomenon.

---

## [Reviewer Report]

This article reviews the formation of coastal extreme waves in water with variable depth. It is clearly written and of interest to the community. One of the important ideas discussed is modulation instability, but it’s not defined. It should be explained in more detail.

This article covers a lot of

---

## [Editor Report]

Dear authors,

I have now received the review comments on your manuscript. All reviewers note the quality of the work and the clarity of its presentation. However, they have made suggestions for improvements to the manuscript. I believe that these can be addressed through a minor revision. 

One of the reviewers provides some clarifications and additional references; the authors should be able to address these comments easily. The other review comments concern the modulational instability (MI). One recommendation is that MI is more clearly defined, whether through a detailed explanation or a brief explanation supported by appropriate citations. The more substantive comments concern the kh = 1.36 threshold for MI and the validity of this threshold beyond idealised long-crested limiting conditions, and a clarification regarding the finding of the theoretical work by Zeng and Trulsen (2012). These comments should be addressed before the manuscript can be accepted, although this can be done within a minor revision. 

One reviewer also questions whether the title should concern “rogue” or “extreme” waves. I invite the authors to consider their suggestion.

---

## [Reviewer Report]

I am satisfied with the revised version of this manuscript. In my opinion, this is an interesting contribution to the literature and I endorse publication.

---

## [Reviewer Report]

I thank the authors for addressing my concerns and I do not have any problems with this moving forward to publication, assuming the other reviewers are happy.

---

## [Editor Report]

I thank the authors for providing detailed responses to each of the review comments on the original manuscript. The reviewers are now satisfied that their suggestions have now been addressed, so I am pleased to accept this paper for publication.